# Cytokines in Systemic Lupus Erythematosus—Focus on TNF-α and IL-17

**DOI:** 10.3390/ijms241914413

**Published:** 2023-09-22

**Authors:** Patricia Richter, Luana Andreea Macovei, Ioana Ruxandra Mihai, Anca Cardoneanu, Maria Alexandra Burlui, Elena Rezus

**Affiliations:** Department of Rheumatology and Rehabilitation, “Grigore T. Popa” University of Medicine and Pharmacy, 700115 Iasi, Romania; patricia.richter@umfiasi.ro (P.R.); ioana-ruxandra_mihai@umfiasi.ro (I.R.M.); anca.cardoneanu@umfiasi.ro (A.C.); maria-alexandra.burlui@umfiasi.ro (M.A.B.); elena.rezus@umfiasi.ro (E.R.)

**Keywords:** systemic lupus erythematosus, TNF-α, IL-17, cytokines, disease activity

## Abstract

Systemic lupus erythematosus (SLE) is an autoimmune disorder known for its complex pathogenesis, in which cytokines play an essential role. It seems that the modulation of these cytokines may impact disease progression, being considered potential biomarkers. Thus, TNF (tumor necrosis factor)-α and IL (interleukin)-17 are molecules of great interest in SLE. TNF-α plays a dual role in SLE, with both immunosuppressive and proinflammatory functions. The role of IL-17 is clearly described in the pathogenesis of SLE, having a close association with IL-23 in stimulating the inflammatory response and consecutive tissue destruction. It appears that patients with elevated levels of these cytokines are associated with high disease activity expressed by the SLE disease activity index (SLEDAI) score, although some studies do not confirm this association. However, TNF-α and IL-17 are found in increased titers in lupus patients compared to the general population. Whether inhibition of these cytokines would lead to effective treatment is under discussion. In the case of anti-TNF-α therapies in SLE, the possibility of ATIL (anti-TNF-induced lupus) is a serious concern that limits their use. The use of anti-IL-17 therapies in SLE is a promising option, but not yet approved. Future studies of these cytokines in large cohorts will provide valuable information for the management of SLE.

## 1. Introduction

SLE is an autoimmune disease based on the altered interaction between the innate and adaptive immune systems. Traditional hypotheses for the immunopathology of SLE include immune complex deposition in tissues and organs, followed by damage caused by autoantibody-mediated reactions [1]. The synthesis of autoantibodies by B cells and the activation of autoreactive T cells constitute the main players in disease pathogenesis. This process leads to the clinical symptoms of disease and organ dysfunction [2]. 

From a clinical perspective, the cause of SLE remains uncertain. There is evidence suggesting that various factors, including environmental elements, genetic predisposition, and estrogen, exert varying degrees of influence on the disease. Most frequently, SLE affects women, being more prevalent in young people and adults. 

There is epidemiological evidence linking SLE to various environmental factors, such as crystalline silica exposure, alcohol consumption (with an inverse association), cigarette smoking, and the use of exogenous estrogens like oral contraceptives and postmenopausal hormones. Additionally, potential associations exist with other external factors like UV radiation, solvents, and pesticides. SLE may also be associated with intrinsic factors like birthweight, reproductive history, endometriosis, and latent infections (Epstein-Barr virus); these latter factors may reflect an indirect role of environmental exposure. 

Epigenetic changes caused by increased oxidative stress, systemic inflammation, high levels of inflammatory cytokines, and hormonal changes represent mechanisms that connect environmental exposure and SLE [3,4].

SLE is characterized by the upregulation of type I interferons (IFN α and ß, primarily released by dendritic cells) and type II interferon (IFN γ, primarily secreted by T cells). This leads to the hallmark of SLE known as the “IFN signature”, which involves elevated levels of IFN-α and increased expression of type I IFN-regulated genes [5]. In SLE patients, the IFN-α levels rise in accordance with disease activity [6]. 

Toll-like receptors (TLR) are pattern recognition receptors (PRRs) that play a fundamental role in the innate immune system. These receptors are significantly expressed in the dendritic cells (DC) of SLE patients. They recognize and bind to bacterial and viral antigens and to other stimuli released during processes like apoptosis and NETosis. In SLE, abnormal TLR activation triggers the production of proinflammatory cytokines and type I IFN. This leads to the synthesis of IFN γ and other interleukins [6,7]. TLR7 and TLR9 are particularly involved in identifying self-nucleic acids and triggering the autoimmune response in SLE [8].

For example, the anti-microbial peptide LL37 may bind to double-stranded DNA (dsDNA) to activate TLR-9 signaling in plasmacytoid DCs (pDC), which may cause excessive IFN production [9].

Additionally, B cells in SLE patients express high levels of TLR, which interact with self-nucleic acids such as DNA, RNA, and other bacterial components. This interaction triggers B-cell class switching and the production of autoantibodies. Thus, these receptors have a significant contribution to the development of SLE through the activation of lymphocytes [6]. 

Chromatin fibers released by dying neutrophils are known as neutrophil extracellular traps (NETs).

The term “NETosis” refers to the death of neutrophils followed by the formation of NETs. Activated neutrophils release reactive oxygen species (ROS). NETs play a crucial role in capturing and eliminating pathogens; they use antimicrobial enzymes such as myeloperoxidase (MPO), neutrophil elastase, cathelicidin, histones, and DNA to exert their antimicrobial function. Excessive NET formation can lead to the release of self-antigens, such as DNA and histones, which can stimulate an autoimmune response. In SLE, this can contribute to the production of autoantibodies and immune complex formation, leading to tissue damage and inflammation [10,11,12].

Cytokines are soluble mediators that play a crucial role in immune response. Thus, the dysregulation of homeostatic cytokine levels has been implicated in the etiology of autoimmune disorders [13,14]. Various innate and adaptive immune systems cells produce these low-weight soluble proteins [1]. Binding to cell surface receptors mediates immune-system activation or functional regulation. They are crucial for developing, differentiating, and activating different immune cells [1,15].

Interleukin (IL)-6, IL-10, IL-17, interferon-alpha (INF-α), and tumor necrosis factor-alpha (TNF-α) are among the essential cytokines that can operate as biomarkers to evaluate disease activity and disease severity. Therefore, modulating these cytokines may constitute a therapeutic approach to managing SLE [1].

There is growing evidence that TNF-α is involved in the pathophysiology of SLE [16]. TNF-α is a cytokine that may play a dual role in SLE; thus, its use is still debated. While it may function as an immunosuppressive mediator that is consistently produced and involved in the development, differentiation, and regulation of immune cells as part of a defense mechanism, it may also act as a proinflammatory mediator released in local tissues during active disease [17,18,19]. 

Patients with SLE have higher serum TNF-α levels, which correlate with disease activity and many systemic manifestations, such as SLE-related cardiovascular disease and lupus nephritis [16]. Further findings indicate that TNF-α expression in SLE may have a “dose-like effect”: aggravation of the condition being correlated with elevated cytokine concentration [20]. However, an anti-TNF-α treatment in SLE is questionable due to the development of antinuclear antibodies (ANA), anti-double-stranded DNA (anti-dsDNA) antibodies, and anticardiolipin antibodies [1]. This research aims to learn more about how TNF-α contributes to the pathogenesis of SLE and to discuss the consequences of anti-TNF-α medication in SLE.

The proinflammatory cytokine IL-17 is involved in the pathogenesis of autoimmune rheumatic diseases, including SLE [1]. A potential implication of IL-17 in the development and disease activity of SLE has been suggested by the evidence that SLE patients reported higher circulating IL-17 levels than healthy controls [21]. Moreover, it is abundant in SLE patients and lupus-prone mice and positively correlated with disease activity. Additionally, IL-17 contributes to the development of SLE comorbidities [1]. In this article, we aim to describe the mechanism of action and implications of IL-17 in SLE.

## 2. TNF-α

The human TNF-α gene is found within the human leukocyte antigen (HLA) class II region on the short arm of chromosome 6 between 6p21.1 and 6p21.3 [16,22].

TNF-α is produced by various immune cells, primarily macrophages, lymphocytes, cutaneous mast cells, eosinophils, and natural killer cells [20,23]. Bacterial lipopolysaccharide (LPS), endotoxin, viral antigens, immune complexes, IL-1, and TNF-α itself can all trigger the release of TNF-α; additionally, some pathophysiological circumstances (inflammation, trauma, heart failure) can stimulate its production [16,24,25].

Following the activation of macrophages and dendritic cells, TNF-α is soluble and expressed on the cell surface as a trimer [1,15].

### 2.1. TNF-α Receptors

To exert its biological functions, TNF-α interacts with two membrane-dependent receptors, TNF receptor 1 (TNFR1) and TNFR2. TNF-α exhibits strong binding affinity to both receptors, and the differential interaction with these receptors is linked to specific functions. TNFR2 displays a reduced affinity for TNF-α compared to TNFR1, suggesting that TNFR2 can bind and then be released, potentially contributing to the amplification or synergy of TNFR1 signaling [16]. 

Moreover, the anti-apoptotic effect induced by downregulation of TRADD, FADD in peripheral blood mononuclear cells (PBMCs) from patients with SLE has been demonstrated, while the TNFR1–TRADD–FADD system may lead to apoptotic signaling [26]. Thus, decreased levels of TRADD and FADD in PBMCs may inhibit the usual apoptotic signaling pathway initiated by TNFR1.

These receptors belong to the TNF receptor superfamily and are membrane glycoproteins [16]. 

Numerous cell types have been identified which express TNFR1, but only a small subset of cells, including T lymphocytes and endothelial cells, have been found to express TNFR2 [27]. Various biological effects of TNF-α, including cytotoxicity and proliferation, are driven by the activation of TNFR1. Depending on the cell’s nature and the surrounding conditions, the activation of TNFR1 promotes a range of cellular responses, such as the stimulation of proliferative processes, apoptosis, or necroptosis [16,28]. 

These two receptors have a highly similar extracellular domain, composed of many cysteine-rich regions implicated in ligand binding. Since the intracellular domains are distinct, they can activate many cytosolic proteins through various signaling pathways [16]. 

As a result of TNF interaction with TNFR1, two distinct TNF receptor signaling complexes (complex I and complex II) are formed. In contrast to complex II, which promotes apoptosis or necroptosis, complex I increases the production of anti-apoptotic genes, which prevent cell death processes primarily via activating transcription factors like nuclear factor κB (NF-κB) [16].

TNFR2 signaling pathways are less fully characterized than TNFR1 signaling pathways. TNFR2 cannot cause cell death directly since it lacks the death domain (DD). TNFR2 involvement significantly increases cell stimulation, migration, and proliferation in contrast to TNFR1 actions, which can trigger inflammation or apoptotic responses (Figure 1). 

When TNFR-associated factor 2 (TRAF2) binds to TNFR2, both the conventional and non-canonical NF-κB signaling pathways are activated [16]. 

### 2.2. TNF-α Mechanisms of Action

TNF-α exerts its role in physiological and pathological circumstances. Among the main actions, it can be mentioned as a significant contributor to the development and function regulation of B cells, T lymphocytes, and dendritic cells, where it acts as an immune regulator, a potent inflammatory mediator that controls the development and persistence of inflammatory processes, but also apoptotic inducer, causing cell death. Thus, depending on the underlying circumstances, TNF-α exhibits pro- and anti-apoptotic effects [16].

The cytokine has two different functions: it fights against infections and leads to pathological consequences. This might occur due to the activation of multiple signaling pathways involved in various biological reactions, including differentiation, proliferation, and cell death [16]. During acute inflammation, TNF-α exerts its role in acute phase reaction, causing the release of CRP and other acute phase reactants [20,29].

TNF-α also acts as an immunomodulatory molecule, stimulating cells of the innate immune system. It can induce fever on the hypothalamus–pituitary axis in conjunction with IL-1 and IL-6 [20].

A difference between short-term and long-term TNF-α exposure in T cells has been detected. Short-term stimulation of activated T lymphocytes with TNF-α causes more T-cell activation, proliferation, and IFN-γ production. At the same time, long-term TNF-α exposure can lead to a reversible loss of the surface T-cell receptor complex, which causes T cells to be less sensitive. IL-2-mediated cell proliferation is unaltered [1,30,31,32].

TNF-α controls various processes, particularly the regulation of vascular adhesion and major histocompatibility complex (MHC) molecules and the activation of other cytokines [1]. TNF is also a B lymphocyte growth factor that triggers the production of IL-1 and IL-6 [1,33].

TNF-α dysregulation induces tissue damage that characterizes the systemic manifestations of SLE. It also contributes to lymphocyte death and defective clearance of apoptotic cells, which further leads to self-antigen presentation with, consequently, autoantibody formation [16,34]. In SLE patients, the increased TNF-α-induced apoptosis raises the number of autoantibodies, increasing autoimmune responses [16]. Although its place in other autoimmune diseases such as rheumatoid arthritis (RA), inflammatory bowel disease (IBD), psoriatic arthritis (PsA), and multiple sclerosis (MS) is well defined, its role in SLE remains to be investigated [16,34].

### 2.3. Genetic Involvement of TNF-α in SLE

From the genetic perspective, various studies have shown a connection between TNF-α gene polymorphism and susceptibility to SLE. 

Additional research has confirmed a significant genetic link between TNF-α gene polymorphism and susceptibility to SLE. For instance, a study examined 204 Indian female SLE patients and age- and sex-matched healthy controls to find two unique single nucleotide polymorphisms (SNPs), G-238A and G-308A, in the promotor upstream of the TNF gene on human chromosome 6 that is linked to a higher risk of developing SLE. According to previous studies, these two alleles correlate to higher TNF-α mRNA transcription [20,35]. 

Polymorphisms may influence the genetic predisposition to SLE in the TNFR2 gene. According to a genotyping investigation, one 196R allele was sufficient in the Japanese population to cause SLE susceptibility [16,36].

On the other hand, some investigations failed to find any connection between SLE and polymorphisms in the TNFR2 gene [16,37,38,39].

### 2.4. TNF-α Implications in SLE Animal Models

TNF-competent New Zealand Black (NZB) mice presented an autoimmune phenotype; on the other hand, TNF-deficient NZB mice also developed a severe lupus-like disease.

An interesting finding was the reversibility of disease when recombinant TNF was administered. Moreover, when TNF was administered early to NZB/White (NZB/W) mice, the onset of lupus nephritis and autoantibodies was delayed [16,40,41,42]. 

In conclusion, when TNF is present and active, it can lead to autoimmune symptoms similar to SLE. However, when TNF is not expressed, it can also result in SLE-like disease. This complexity emphasizes the multifaceted role of TNF in the development of SLE and the importance of additional research in order to determine its precise contribution to the disease.

### 2.5. TNF-α Levels in SLE

TNF-α involvement in the pathogenesis of SLE is unclear; some researchers have discovered that it increases SLE susceptibility, while others have documented how it protects SLE patients (Table 1).

Patients with SLE with active disease have been found to have higher serum levels of TNF-α and its soluble receptors than SLE patients with inactive disease [16,43,44]. In SLE patients, elevated TNF-α levels have been linked to the severity of the disease [16].

Nevertheless, other researchers have discovered lower levels of TNF-α in SLE patients, especially those with severe disease [16]. TNF-α levels were higher in patients with inactive disease than in patients with very active disease and controls, suggesting that TNF-α may also contribute as a protective factor in SLE patients [1,18,45].

SLE patients also had higher TNF-α levels in plasma and inflamed kidneys [16,46]. 

In the serum of SLE patients, TNFR1 and TNFR2 expression levels were significantly increased [16,46].

Patients with SLE showed significantly decreased levels of TNF adaptor proteins, TNF receptor-associated death domain (TRADD), Fas-associated death domain (FADD), TRAF2, and receptor-interacting protein kinase 1 (RIPK1) expressed in their peripheral blood mononuclear cells, according to Zhu et al. These levels were shown to be inversely associated to the SLE activity index [16,26].

**Table 1 ijms-24-14413-t001:** The relation between circulating TNF-α levels and disease activity in SLE patients.

Study	Number of Subjects	Measure of Active SLE	TNF-α Levels in SLE Patients vs. HC	TNF-α Levels and Disease Activity	Other Findings
[43]	40 SLE 24 HC	SLEDAI > 7 ECLAM	Significantly higher serum levels of TNF-α in patients with active disease than in patients with inactive disease	No significant difference between levels of TNF-α in patients with inactive disease and HC	Strong association between serum TNF-α levels and anti-dsDNA, complement C3, C4 and CH50
[47]	208 SLE	SLAM ≥ 7	-	Significant, but weaker correlation of circulating TNF-α with disease activity	The increased activity of the TNF-α was strongly linked to the cardiovascular and renal complications in SLE; conversely, there was an negative correlation between dermatologic manifestations and TNF-α activity.
[18]	52 SLE 25 HC	SLEDAI ≥ 13 (very active) SLEDAI = 3–12 (moderately active)	Significantly higher level of TNF-α in SLE patients than in HC	Higher levels of TNF-α in patients with inactive SLE compared to patients with very active disease	Elevated TNF-α levels were noted among individuals with a prior history of infection, although the difference was not statistically significant.
[48]	40 SLE 10 HC	SLEDAI	Significantly increased level of serum TNF-α vs. HC	Significant correlation of TNF-α with the SLEDAI score	Significant elevation of serum TNF-α in patients with vasculitis compared to patients without vasculitis
[49]	40 SLE 20 HC	SLEDAI ≥ 10	Significantly higher level of TNF-α in SLE patients when compared to HC	Statistically significant higher TNF-α level in SLE group with active vs inactive disease	Positive association between serum TNF- α, IL-6 and SLEDAI
[50]	61 SLE	SLEDAI ≥ 5	Significantly elevated TNF-α levels in SLE patients classified as having pulmonary phenotype vs non-pulmonary phenotype	No correlation between serum TNF-α and SLEDAI score in both phenotypes	Significantly higher levels of TNF-α in SLE patients with the restrictive pattern of pulmonary involvement vs obstructive pattern
[51]	653 SLE 62 HC	-	High serum TNF-α levels in SLE patients compared to HC	-	Positive association between serum TNF-α and IFN-α levels
[52]	82 SLE 14 HC	SLEDAI	Significantly higher level of TNF-α in SLE patients vs. HC	Higher levels of TNF-α in SLE patients with active disease than patients with inactive disease, but no significant differences between the groups	-
[44]	45 SLE 20 HC	SELENA-SLEDAI	Significantly higher serum TNF-α concentrations in SLE patients compared to HC	Weak correlation of TNF-α with the SLEDAI in SLE patients	High levels of TNF-α in SLE patients with renal involvement
[53]	437 SLE 322 non-SLE controls	SLEDAI-2K SLAM PtGDA	Significantly increased TNF-α in SLE patients vs. matched HC	Strongest correlations to all three measurements of disease activity (SLEDAI-2K, SLAM, PtGDA) with TNF-α	Particular association of TNF-α with active nephritis
[4]	68 SLE 60 HC	SLEDAI ≥ 5	Higher levels of serum TNF-α in the SLE groups vs. HC	● Higher serum TNF-α levels in SLE patients with severe disease activity than in patients with moderate or mild SLE activity; ● Higher serum TNF-α levels in moderate vs mild SLE activity	Positive association between anti-dsDNA antibody and TNF-α levels

HC = healthy controls; SLEDAI-2k = SLE Disease Activity Index 2000; SELENA = Safety of Estrogens in Lupus Erythematosus National Assessment; PtGDA = Patients’ assessment of Global Disease Activity; dsDNA = double-stranded DNA antibody; ECLAM = European Consensus Lupus Activity Measurement; SLAM = Systemic Lupus Activity Measure.

### 2.6. TNF-α and Clinical Manifestations in SLE

In individuals with rheumatic disorders, including RA, spondyloarthropathies (SpA), or SLE, TNF-α is overexpressed. Regarding SLE, higher serum and gene expression levels of TNF-α have been found. Moreover, these levels positively correlate with disease activity, especially in patients with renal involvement [17].

Globally, cardiovascular illnesses are among the main causes of death, many of them being the consequence of the appearance and progression of atherosclerotic lesions [54], which justifies their multidisciplinary approach, especially in the context of rheumatologically involvement [55]. In this sense, Svenungsson et al. focused on the association between inflammation, dyslipoproteinemia, and cardiovascular disease in SLE patients, revealing high triglyceride and low HDL (high-density lipoprotein) cholesterol levels as disease activity markers and elevated levels of TNF-α/TNFR [16,47,56].

TNF-α may also be involved in the cutaneous development of subacute cutaneous lupus erythematosus (SCLE) since investigations on refractory skin lesions biopsied from SCLE patients revealed a substantially positive epidermal distribution of TNF-α compared to control skin lesions [20,57].

SLE nephritis is a prototype of immune-complex-induced renal disease [1,58]. All forms of lupus nephritis were discovered to have high levels of TNF-α in the glomeruli, and the expression of TNF-α was linked with renal inflammatory activity [30]. High TNF-α levels have also been reported in non-SLE nephropathies, such as membranous nephropathies and nephritic syndrome [59]. These results support the idea that TNF-α pathogenic involvement is in developing or maintaining glomerular barrier dysfunction in renal disorders [1].

Several studies have suggested that cellular immunity may play a role in developing SLE; proteinuria is thought to be caused by the deposition of immunoglobulins and complement system components on the epithelial side of the glomerular basement membrane. Evidence of elevated TNF-α expression in the glomeruli, high urine levels, and complement cascade activation contribute to this role [1].

### 2.7. Anti-TNF-α Therapies in SLE

The impact of TNF-α-based therapy in SLE is controversial and can differ due to the dual role of TNF-α as a mediator of inflammation and a regulator of autoimmunity [17].

It was interesting to assume that anti-TNF-α monoclonals, which can suppress immunological mechanisms, would be helpful in SLE treatment. Although some studies were not as encouraging, there was some initial expectation that this treatment would be successful [17]. Contrastingly, other studies show that anti-TNF-α medication is effective and successful for connective tissue disease (CTD) [60], particularly SLE and cutaneous lupus erythematosus (CLE) [20].

The two TNF-α blockers that have been most commonly investigated are infliximab and etanercept [17].

Because of its chimeric structure, infliximab is the anti-TNF-α molecule with the highest level of immunogenicity. However, studies have shown that infliximab has a good safety and tolerability profile in SLE patients. In particular, infliximab short-term induction therapy with azathioprine or methotrexate induced long-term improvement in lupus nephritis [16,61,62].

Etanercept has also been investigated in clinical trials, including a randomized, double-blind, phase II, multi-center study for the treatment of lupus nephritis (NCT00447265) and two phase II open-label trials for the treatment of discoid lupus erythematosus (DLE), even though the Food and Drug Administration (FDA) has not yet approved it for the treatment of SLE (NCT02656082 and NCT00797784) [16,63,64,65]. Long-term etanercept therapy for refractory lupus arthritis was found to be relatively safe and effective in an observational trial [16].

Another study that investigated patients treated with adalimumab and etanercept in SLE showed that the median prednisone dose was considerably lowered from 15 mg/day to 5 mg/day during the observation period [17,66]. These findings contribute to the idea that anti-TNF-α medication may effectively manage refractory lupus arthritis [17]. 

Etanercept has also been demonstrated to be effective in treating rhupus, which shares characteristics with both RA and SLE [16,66,67].

Etanercept, plasmapheresis, and high-dose intravenous gamma globulin were successfully used to treat pregnant SLE patients with severe diffuse proliferative nephritis [16,68].

A case report study demonstrated that etanercept could improve clinical symptoms and overall quality of life in people with SCLE [16,69].

#### The Negative Impact of Anti-TNFα Therapies in SLE

TNF-α blockers have been associated with developing ANA and anti-dsDNA antibodies. As a result, some individuals may also experience clinical symptoms comparable to those of idiopathic lupus. When this occurs, it is thought to be anti-TNF-α-induced lupus [17]. The medical condition is known as anti-TNF-α-induced lupus erythematosus (ATIL). The diagnosis of ATIL is typically established when there is a different temporal relationship between the development of symptoms and the introduction of anti-TNF-α therapy or an increase in its dosage [16,20].

A study was conducted by Picardo et al. to assess the incidence of ATIL and its clinical and serological characteristics in patients receiving anti-TNF-α drugs. They showed a higher incidence of ATIL in patients receiving infliximab versus adalimumab [17,70].

TNF-α targeting therapies are currently used to treat inflammatory and autoimmune conditions such as RA, IBD, and psoriasis. Despite their clinical success, anti-TNF-α drugs have limited use due to serious side effects and the emergence of ATIL. Current TNF-α blockers likely contribute to these side effects by preventing TNF-α from engaging with the regulatory TNFR2 receptor and the pro-inflammatory and pro-apoptotic TNFR1 receptor. As a result, the TNFR2 signaling regulatory function is lost [16].

Data have shown that nephritis can develop after anti-TNF-α medication therapy [16,71,72]. Drug use targeting TNF-α and TNFRs in SLE is still controversial. Therefore, additional research is necessary to establish the beneficial treatments [16].

## 3. IL-17

The IL-17 family plays a significant role in defense against fungi and bacteria [13,21,73]. Six cytokines constitute the IL-17 family: IL-17A to IL-17F. The first identified cytokine was IL-17A, also termed IL-17, the most deeply investigated. IL-17A is linked to the pathogenesis of various autoimmune conditions such as psoriasis, MS, IBD, and seronegative SpA, as well as animal models of autoimmunity [74,75,76].

### 3.1. Receptors of IL-17

Specific membrane receptors of the IL-17 family (IL-17R) are involved in IL-17 signaling [13]. The IL-17R family is formed of five members, IL-17RA-E. These family members encode two extracellular fibronectin II-like domains and one intracellular SEFIR (similar expression of fibroblast growth factor genes/IL-17 Receptor) domain [77].

The receptor for IL-17A consists of two different subunits: IL-17RA and IL-17RC. When IL-17 binds to its receptor, both the conventional (canonical) and non-canonical pathways become activated (Figure 2). Following the canonical path, when IL-17 binds to the IL17RA/IL17RC complex, it induces the adaptor protein Act1 to attach to the SEFIR complex, which further leads to the ubiquitylation of TRAF6 (TNFR-associated factor 6) and also to the activation of the Mitogen-activated protein kinase (MAPK), CCAAT-Enhancer-Binding protein β/δ (C/EBPβ/δ), and NF-κB pathways [13].

Interestingly, genes encoding inflammatory cytokines and chemokines are upregulated. These genes are also activated by TNF-α interaction with TNF receptors I and II, allowing for a synergistic effect between TNF-α and IL-17 [78].

Additionally, in the non-canonical pathway, IL-17 promotes the phosphorylation of Act1, which inhibits splicing factor 2 (SF2), a factor involved in mRNA destabilization, and this leads to the recruitment of the mRNA stabilizer human antigen R (HuR), which generates the stabilization of the mRNA and the secretion of proinflammatory cytokines and chemokines [13].

### 3.2. Cellular Sources of IL-17

Based on the cytokines they produce, transcription factors, and the many roles they play, CD4+ T cells can be categorized into several subsets, including T helper (Th)1, Th2, Th17 cells, regulatory T cells (Treg), and T follicular helper (TFH) cells [74].

The proinflammatory cytokine, IL-17, is primarily released by activated Th-17 cells [13,21,74]. IL-17 is not exclusively produced by CD4+ Th17 cells; it can also be expressed by activated CD8+ cytotoxic T (Tc) cells, double-negative (DN; CD3+CD4-CD8-) T cells, γδ-T cells, NK cells, dendritic cells, macrophages, and neutrophils [2,13,74,79,80,81,82]. More precisely, Crispín et al. found that DN T cells from SLE patients secrete significant amounts of IL-17 with the same efficacy as CD4+ T cells. It seems that neutrophils and mast cells are able to release IL-17 via extracellular traps [77,83,84].

The presence of additional proinflammatory cytokines, such as IL-21, can enhance the production of IL-17 by memory T cells [84].

Apoptotic material stimulates pDC, which releases type I IFNs, mainly IFN-α. Further, Th-17 secrete IL-17 [21]. As a result, Th-17 cells are stimulated to differentiate, through the release of cytokines by DC, to the detriment of Treg cells [21,77].

After being activated during antigen presentation in secondary lymphoid organs [77], naive CD4+ T cells undergo differentiation into Th1, Th2, and Th17 cells [2]. The activation of immature T cells via the T cell receptor (TCR) and the involvement of costimulatory molecules in the presence of transforming growth factor (TGF)-β, IL-6, IL-21, IL-23, and IL-1β are necessary for the differentiation of naive T cells into Th17-cells in vitro [74,85,86].

Like other Th-cells, Th17 requires certain cytokines and transcription factors for activation and proliferation [87]. Th17 cell development involves three phases: initiation, amplification, and expansion and stabilization. Notably, IL-21 and TGF-β are primarily involved in the first stage, and IL-1β and IL-6 are implicated in the second [78]. High IL-6 production by antigen-presenting cells may cause an unbalanced naive T-cell differentiation, producing Th17 cells rather than Treg cells [84]. To develop their maximum pathogenic potential, IL-23 is essential for the expansion and stabilization of Th17 cells in their final stage [78,87]. Thus, Th17 pathogenicity is amplified by IL-23 exposure [78].

An important factor in autoimmune and inflammatory conditions is the Th17/Treg balance [78]. The literature data suggest that the imbalance between Th17 cells, Th1, or Treg lymphocytes caused by a disturbance in cytokine regulation plays a critical role in the development of SLE [87]. Concerning autoimmunity, Treg inhibits its effect, whereas Th17 stimulates it. The transcriptional regulator FoxP3 and TGF-β are needed for Treg differentiation. On the other hand, the addition of IL-6 to TGF-β promotes Th17 differentiation by suppressing FoxP3 and upregulating RORc [78].

Apart from the signature cytokine IL-17, Th17 cells also produce other important cytokines such as IL-21, IL-22, TNF- α, and the CCR6 chemokine ligand CCL20 [2,13,88,89,90].

As Th17 cells are involved in the pathology of numerous autoimmune diseases, specific molecules that regulate Th17 cells have become more important [87]. There are two ways to activate Th17 cells through some molecules that stimulate or inhibit their differentiation. For instance, IFN-γ and IL-4, cytokines released by Th1 and Th2 cells, are the main inhibitors of Th17 cells, whereas IL-6/TGF- β can stimulate them [87,91]. Abnormal activation of Th17 cells is linked to the pathology of rheumatic diseases such as RA, SpA, and systemic sclerosis (SSc), but also MS, IBD [33,34], and psoriasis [13,92,93,94,95,96,97,98,99,100,101,102].

### 3.3. Roles of IL-17

IL-17, secreted by DN T cells and Th-17 cells, also has a role in developing spontaneous germinal centers. IL-17 cooperates with IL-23, IL-17F, and IL-21 to create a complex network that stimulates inflammatory response and generates tissue damage in SLE. It also triggers the production of other proinflammatory cytokines, such as IL-1 and IL-6. IL-21 contributes to the balance between Th17 and Treg cells, while IL-10 belongs to the IL-2 cytokine family associated with chronic inflammation [21,87].

During active disease, the suppressive function of Treg cells is diminished, accompanied by a decrease in the expression of the transcription factor FoxP3. This reduction in Treg cell function can also be influenced by IFN-α produced by DC. Additionally, the proliferation of Th17 cells is associated with decreased IL-2 production and elevated levels of IL-6 [78].

Under the influence of TGF-β and IL-6, naive T cells develop into Th17 cells. IL-23 is required for pathogenic Th17 cells to differentiate and consequently to present effector functions. Both IL-23 and IL-17 play critical roles in inflammation where they belong to the IL-23/IL-17 axis, being involved in the differentiation and activation of Th17 cells [103].

In order to develop strategies for SLE treatment, there is growing evidence on targeting IL-17A. The primary focus should be on individuals whose disease is driven by IL-17 [78].

Indirect targeting of IL-17 is linked to the inhibition of Th17 cell synthesis. Particularly, blocking cytokine pathways like IL-6, IL-1, or IL-23 prevents the development of Th17 cells [78].

Ongoing studies involve IL-6 inhibitors. IL-6 plays a key role in the generation of Th17 cells, and inhibiting IL-6 may potentially target SLE activity by an indirect IL-17 suppression [78].

Because B cells have pivotal roles in SLE pathogenesis, the upregulation of B lymphocyte stimulator (BLyS) contributes to the development of SLE. BLyS serves as a survival factor for B cells by preventing their apoptosis, stimulating their proliferation differentiation through interaction with IL-17, finally leading to an increase in the production of autoantibodies [103].

Ustekinumab is an inhibitor of IL-12 and IL-23, which are essential for Th17 differentiation [103]. The monoclonal antibody inhibits the Th1 and Th17 pathways. It showed promising results in a double-bind Phase II study for subacute cutaneous lupus and active lupus [103].

In conclusion, larger clinical trials are necessary in order to assess the safety and efficacy of these molecules in SLE.

IL-17 alone or in association with B-cell activating factor (BAFF) controls B cells’ survival, proliferation, and differentiation into immunoglobulin-secreting cells [21]. IL-17 increases B-cell growth and, thus, autoantibody synthesis [87]. Consequently, autoantibodies are formed, immune complexes are accumulated in the target organs, complement is activated, and the risk of tissue damage is increased. Further, these immune complexes induce pDC activation [21].

By increasing the production of matrix metalloproteinases (MMPs) and intercellular adhesion molecule-1 (ICAM-1), IL-17A promotes T-cell activation and infiltration into tissues. IL-17A can boost the immune response by enhancing organ damage in SLE [74].

Additionally, it has been demonstrated that the genetic deletion of IL-17 improves the clinical evolution of SLE [69]. This confirms that IL-17 stimulates immune cell migration to target tissues, perpetuating and amplifying the inflammatory response and thus exacerbating SLE activity.

### 3.4. High Levels of IL-17 and Th17 Cells in SLE

Many human and mouse reports show a dysregulation of IL-17 in SLE. In two different lupus animal models, the levels of IL-17 and IL-17-producing cells were increased [13,104].

Chronic immune-mediated conditions have been associated with higher IL-17 levels [13]. Wong et al. found significantly elevated plasma IL-17 and circulating Th17-cell amounts in SLE patients compared to healthy subjects [105]. Other studies confirmed that patients with SLE had more significant amounts of IL-17-producing cells in their PB [13]. Henriques et al. found increased amounts of Th17 and Tc17 in the PB of SLE patients [106]. In the report by López et al., fresh peripheral blood cells from SLE patients and healthy control (HC) were examined for intracellular IL-17. Following the higher levels of IL-17 found in the serum, the proportion of IL-17+ cells was more significant in both neutrophils and CD4+ lymphocytes from SLE patients than in controls [107].

Additionally, under the inflammatory conditions seen in SLE, IFN-activated neutrophils can release IL-17 [107]. Yang et al. reported a high frequency of circulating Th17 cells in active SLE patients compared to patients with the inactive disease [2,108]. Moreover, most of the IL-17-producing T cells in SLE were restricted to the T-cell subset that expressed high levels of costimulatory molecules CD80 and CD134 [2,109].

Many studies demonstrated that serum IL-17A was notably more elevated in active patients than in controls [2,108,110,111]. Moreover, a positive correlation was seen between IL-17 concentrations and SLEDAI scores [105,112]. Regarding this association between the proinflammatory cytokine and disease activity, Galil et al. showed that IL-17 concentrations and SLEDAI show a positive correlation [113]. Furthermore, IL-17A was positively correlated with RORγt mRNA, erythrocyte sedimentation rate, IgG, IgA, and SLEDAI score in new-onset SLE patients, highlighting the significance of the Th17–IL-17 axis in this disease [2,114]. Interestingly, in some studies, these associations were not seen [115,116] (Table 2).

### 3.5. Clinical Manifestations

SLE is known for its multisystemic involvement, such as the skin, kidney, central nervous system (CNS), or hematological system. Besides the increased values in serum, high levels of IL-17A have been seen mainly in lupus nephritis and SLE patients’ skin and CNS lesions [74].

Many studies show that in the kidneys of lupus nephritis (LN) patients, IL-17 levels and Th17 cells were elevated [2,84,126]. Th17 cell frequency was higher in the LN group than in the SLE without nephritis group and the HC group [127]. IL-17 was detected in high levels in the renal biopsies of LN patients where inflammatory cellular infiltrates were present [109]. In a different study, which included SLE patients without a history of renal disease, subjects with a history of LN in remission, and participants with current LN, the urinary expression of Th17-related cytokines was elevated in all three groups compared to controls [128]. Histological examination of SLE patients with class III/V, IV/V, and V nephritis revealed the presence of IL-17+ T cells infiltrating glomerular and interstitial areas. There was a strong association of IL-17 levels with hematuria and SLEDAI scores in the glomeruli and interstitium [129].

Th17 cells migrate to the kidney, and, thus, they contribute to inflammatory reactions. The number of Th17 cells in peripheral blood and IL-17 levels in the urine of SLE patients had to be determined to look into their associations with disease activity and renal involvement. Patients had considerably higher urinary IL-17 and Th17 expression levels than controls. Moreover, significant correlations between Th17 cell frequencies and IL-17 levels with renal biopsy LN were observed [130]. Furthermore, IL-17+ and DN T cells were discovered in the renal biopsies of patients with lupus nephritis, suggesting that renal inflammation is directly induced by T cells [2,84,131]. These findings were also confirmed by in vitro studies [2,104].

Interestingly, cerebrospinal fluid levels of IL-17 were significantly elevated in SLE patients with CNS infection compared to SLE controls or neuropsychiatric SLE [132].

IL-17A expression was increased in patients with DLE, SLE, and SCLE versus controls. Notably, IL-17A+ T cells in the SCLE lesions were linked to serum anti-Ro antibodies [116].

Compared to active SLE patients without vasculitis, patients with active SLE and vasculitis presented significant amounts of Th17 cells; thus, IL-17 may also be involved in the pathogenesis of vasculitis in active SLE [108].

To evaluate the level of circulating IL-17 as a biomarker of SLE activity, Yin et al. performed a systematic review and meta-analysis. The authors assessed the relationship between the level of circulating IL-17 and disease activity in SLE patients. They found a weak positive association between circulating IL-17 and the presence of SLE. However, SLE patients with active disease expressed higher levels of circulating IL-17 than those with inactive SLE. These introductory remarks support the clinical relevance of IL-17 in SLE treatment. Additional randomized controlled trials are required [21].

### 3.6. Anti-IL-17 Treatment

Psoriasis, PsA, and ankylosing spondylitis can all be effectively treated with agents that target IL-17, highlighting the cytokine’s significance in developing these autoinflammatory disorders. Since anti-IL-17 therapy has successfully treated other autoimmune diseases, this supports the possibility of treating SLE to reduce disease activity [74].

Increased IL-17A production is crucial in developing germinal centers (GCs) in the spleen of BXD2 mice. The formation of GCs depends on interactions between CD4+ T cells and B cells, which become impaired when IL-17 signaling is blocked [133].

Study SELUNE (NCT04181762), a phase III randomized, double-blind trial, aims to investigate the efficacy and safety of secukinumab, an anti-IL-17A monoclonal antibody, in combination with the standard of care therapy in patients with active lupus nephritis. No results are available yet [134].

## 4. Conclusions

Anti-TNF-α therapy for SLE patients is still a highly debatable and intriguing option. Unfortunately, data from the literature reveal that there is still insufficient evidence to support the administration of anti-TNF-α drugs in SLE. In addition, these drugs carry a risk of drug-induced CTD. ANAs, which are linked to several primary autoimmune conditions and CTDs, are frequently generated due to anti-TNF-α therapy. Due to these concerns, there has been an ongoing debate concerning the use of anti-TNF-α therapy in treating primary CTDs. TNF-α-blocking drugs are currently only used in refractory cases of SLE and are not indicated as the first line of treatment.

IL-17A is considered a promising therapeutic target for SLE. The importance of the IL-17A axis role in the pathophysiology of SLE is now known. It should be taken into account to develop new techniques to inhibit IL-17A signaling pathways.

## Figures and Tables

**Figure 1 ijms-24-14413-f001:**
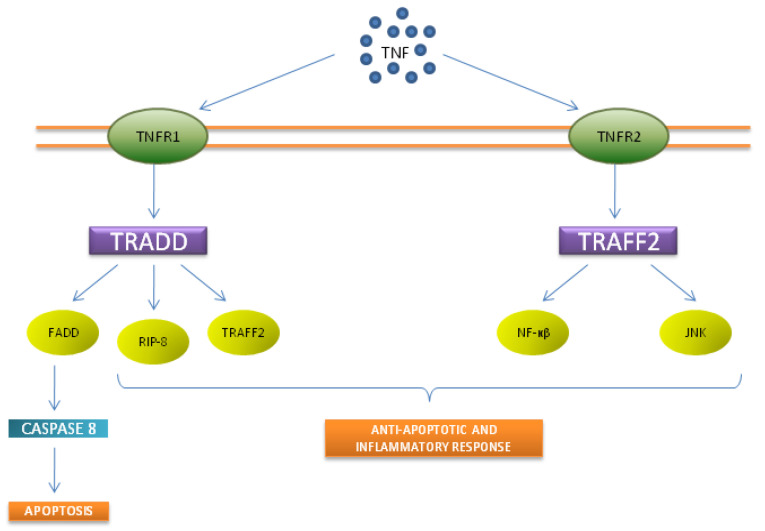
The two surface TNF receptors are called TNFR1 and TNFR2. The death domain (DD) in the cytoplasmic tail is a distinctive feature of TNFR1. Firstly, TNR1 binds to TNFR-associated death domain (TRADD). The newly created complex functions as a platform where different mediators such as receptor-interacting protein 1 (RIP-8), Fas-associated death domain (FADD), and TNF receptor-associated factor-2 (TRAFF2) are recruited. Subsequently, TNFR1 attaching to FADD and TRADD results in caspase eight recruitment, oligomerization, and activation. Finally, apoptosis is induced. Contrary to TNFR1, TNFR2 has no death domain. Yet it still recruits some adaptor proteins, interacting directly with TRAFF2. By activating transcription factors including NF-κB and stress-activated JNK (c-Jun N-terminal Kinase), TRAFF2 promotes inflammatory responses.

**Figure 2 ijms-24-14413-f002:**
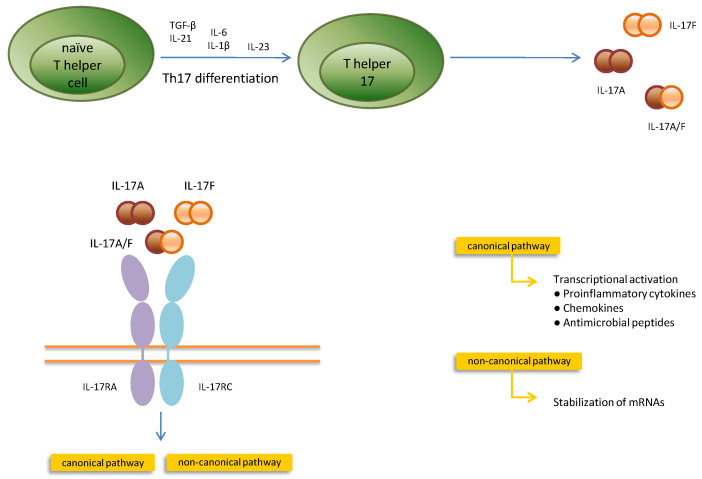
The development of Th17 cells requires three stages that involve different cytokines: the initiation (IL-21 and TGF-β), the amplification (IL-1 β and IL-6), and a phase of expansion and stabilization (IL-23). Th17 cells are the primary source of IL-17A, IL-17F, and IL-17A/F. The homodimers IL-17A and IL-17F and the heterodimer IL-17A/F bind to the same receptor composed of IL-17RA and IL-17RC subunits. After IL-17 attaches to its receptor, two different pathways, the canonical and the non-canonical, are activated. The canonical pathway contributes to the synthesis of antimicrobial peptides and proinflammatory cytokines and chemokines, whereas the non-canonical way induces the stabilization of messenger RNA encoding for inflammatory mediators.

**Table 2 ijms-24-14413-t002:** Circulating IL-17 levels in SLE patients.

Study	Number of Subjects	Measure of Active SLE	IL-17 Levels in SLE Patients vs. HC	IL-17 Levels and Disease Activity	Other Findings
[111]	36 SLE 18 HC	NA	Significantly elevated plasma levels of IL-17 in SLE patients vs. HC	No significant positive correlation between IL-17/ IL-4 ratio and SLEDAI score	No correlation in terms of treatment
[105]	80 SLE 40 HC	SLEDAI ≥ 6	Significantly higher levels of IL-17 in SLE patients than controls	Positive and significant association between IL-17 and SLEDAI score	Positive correlation between IL-17 levels and CXCL10 concentration in SLE patients
[114]	60 new-onset SLE 56 HC	SLEDAI	Higher IL-17A value in SLE patients compared to HC	Positive correlation of IL-17 with SLEDAI score	-
[116]	26 DLE 23 SLE 17 SCLE 13 HC 10 psoriasis	SLEDAI	● Significantly greater serum IL-17A concentrations in DLE and SLE patients than in HC ● Similar serum IL-17A levels in SCLE and HC	No correlations between SLEDAI and serum IL-17A	No correlation of IL-17A with the corticoid or hydroxycloroquine therapy
[110]	70 SLE 36 HC	SLEDAI > 6	Elevated serum levels of IL-17 in SLE patients vs. HC	No correlation between SLEDAI and serum levels of IL-17	● Elevated serum levels of IL-6 and IL-10 in SLE patients vs. HC ● Among the cytokines studied, only IL-10 demonstrated a significant correlation with SLEDAI
[117]	15 active SLE 15 inactive SLE 15 HC	SLEDAI ≥ 6	Significantly higher serum levels of IL-17 in active and inactive SLE patients compared to HC	● Positive correlation of serum IL-17 with SLEDAI score	-
[118]	40 pSLE 20 HC	SLEDAI ≥ 10	Significant serum levels of IL-17 in pSLE than in HC	Statistically significant correlation between the levels of serum IL-17 and SLEDAI	Higher IL-17 levels in pSLE patients with cutaneous and haematological manifestations
[115]	98 SLE 39 HC	SLEDAI-2k ≥ 4	Significantly higher serum IL-17 levels in SLE patients vs. HC	No correlation between IL-17 and SLEDAI-2k	● Positive association of IL-17/IL-6 ratio with SLEDAI-2k ● Significantly elevated levels of IL-17 in SLE patients with CNS disease ● Notable correlation between serum IL-17 and IL-6
[113]	72 SLE 70 HC	SLEDAI-2k > 4	Significantly elevated serum levels of IL-17 in SLE patients vs. HC	Correlation between IL-17 and SLEDAI	● Significantly elevated levels of IL-6 in SLE patients vs. HC
[119]	60 SLE 24 RA 24 HC	SLEDAI ≥ 6 32 active SLE 28 inactive SLE	Significantly elevated levels of IL-17 in SLE vs. HC	IL-17 significantly higher in SLE patients with active vs. inactive disease	● No statistically significant association between IL-17 and the medications administered to SLE patients. ● As well as IL-17, IL-10 and IL-6 also showed a significant increase in SLE patients with active disease compared to those with inactive disease.
[120]	57 SLE 42 HC	SLEDAI	Significantly higher median levels of IL-17 in SLE patients compared to controls	-	In SLE patients, low vitamin D levels were linked to elevated serum concentrations of IL-17 and IL-23.
[121]	111 SLE 80 HC	-	Significantly higher expression of IL-17 in SLE patients vs. HC	-	Increased expression of IL-1β, IL-6, IL-8, IL-10, IL-17, IFN-γ in SLE
[99]	32 SLE 20 HC	SLEDAI-2k	Significantly higher IL-17 levels in SLE patients vs. HC	Significant positive correlation between IL-17 levels and SLEDAI-2K	Higher IL-17 levels in lupus nephritis than in SLE with no renal involvement
[122]	50 SLE 39 HC	SLEDAI ≥ 6	High levels of IL-17 in SLE subjects vs. HC	Significant positive correlations of IL-17 levels with SLEDAI	No significant associations between LN, non-LN patients, and HC
[123]	120 SLE	Modified SLEDAI-2k ≥ 1 BILAG 2004	-	Positive association between serum IL-17A with total scores of BILAG 2004 and Modified SLEDAI-2K scores	Correlation between urine IL-17A with SLE disease activity
[124]	68 SLE	MEX-SLEDAI 34 active SLE 34 inactive SLE	-	No significant correlations of IL-17 between active and inactive SLE patients	Significant positive association between IL-17 and IL-10
[125]	50 JO-SLE 25 HC	SLEDAI-2k	Significantly higher plasma levels of IL-17 in JO-SLE patients compared to HC	No relationships between IL-17 levels with the disease activity of SLE patients	No correlation between 25-OH vitamin D and IL-17 in these patients

HC = healthy controls; vs = versus; IL = interleukin; SLEDAI = SLE disease activity index; RA = rheumatoid arthritis; CNS = central nervous system; pSLE = pediatric SLE; DLE = discoid chronic lupus; SCLE = subacute lupus erythematosus; JO = juvenile onset; MEX-SLEDAI = Mexican Systemic Lupus Erythematosus Disease Activity; BILAG = British Isles Lupus Assessment Group’s 2004 index; LN = lupus nephritis.

## Data Availability

Not applicable.

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
