# Peer review of "Cytokines in Systemic Lupus Erythematosus—Focus on TNF-α and IL-17"

_ijms, 2023, doi:10.3390/ijms241914413_

Round 1

Reviewer 1 Report

See the attachment, please.

The comments are included in the main review file.

Author Response

Thank you for your help and understanding!

Reviewer 2 Report

In order for an acceptance of the work, I suggest same corrections to the authors

Line 29: SLE is an autoimmune disease based on the interaction between the innate and adaptive immune system, not correct. 

It is better:  SLE is an autoimmune disease based on the altered interaction between the innate and adaptive immune system.

Line 29: Broaden the introduction and general information about SLE, those reported are limited. The authors don’t speak about the role of TLRs, DCs, the IFN-I signature, NETosis  that characterizes the disease, and not speak about any  hypotheses present in literature on how tolerance to self is broken in the disease and of molecules capable of breaking self-tolerance in association with DNA (for example LL37). The multifactorial nature of SLE, due to genetic and predisposing factors but also to external factors are not considerate clearly. The abnormal production of TNF-α and IL-17 in SLE depend on the upstream mechanisms, which should be mentioned  for better understanding the disease  and to better contextualizing  the cytokines under studies.

Line 79 and Fig 1:  TNF receptors: describe better or do not put idem for line329 and figure 2

Line 186 to 195: In mice anti TNF drugs, has been shown different finding or good outcomes? It is not clear (contradiction). TNF administration has good effects in mice Lupus model, as reported by authors (conytradiction). Moreover, in SLE mice model there are also publications Contrary to anti TNF  findings.  Add data and references of different findings and write 2.4 more clearly, it not clear and contradictory 

Line 200 and 220: Introduce more update bibliography if possible

Line 422. i suggedt to Add hypothesis about T reg can convert to IL 17 like cell under infiammatory condition exspecially the presence of IL6. It is possible that Th 17 cells could be diverted T reg cells. Anti IL 17 and IL 23 treatments could restore their function.

Author Response

(The authors gave the same response as above.)

Round 2

Reviewer 1 Report

Dear Authors,

In my opiniom, the changes made manuscript much better.

Reviewer 2 Report

Now the work is good